# Impact of *MED12* mutation and CDK8 activity on uterine leiomyoma growth and response to gonadotropin-releasing hormone agonist treatment

**Saki Tanioka**[1], **Ryoko Asano**[1,2*], **Yukihide Ota**[1], **Koichi Nagai**[1], **Katsuya Takenaka**[3,4], **Taichi Mizushima**[1], **Yohei Miyagi**[3], **Etsuko Miyagi**[1]

**1** Department of Obstetrics, Gynecology and Molecular Reproductive Science, Yokohama City University Graduate School of Medicine, Yokohama, Japan, **2** Department of Gynecology, Yokohama City University Medical Center, Yokohama, Japan, **3** Molecular Pathology and Genetics Division, Kanagawa Cancer Center Research Institute, Yokohama, Japan, **4** TR Company, Shin Nippon Biomedical Laboratories, Ltd., Kagoshima, Japan

\* new_official_asano@hotmail.co.jp

## Abstract

*MED12* exon 2 mutation is the most frequent mutation associated with uterine leiomyomas. *MED12* wild-type leiomyomas have a higher growth potential than mutant leiomyomas, suggesting that the mutation limits leiomyoma growth. MED12 forms a complex with CDK8 and is involved in the phosphorylation of RNA polymerase II, playing a role in transcriptional regulation. However, its mechanism of action in leiomyoma growth is not clear. We aimed to clarify the relationship between *MED12* mutation status, response to gonadotropin-releasing hormone (GnRH) agonist treatment, and CDK8 activity in leiomyomas. We also examined the effects of CDK8 inhibitors on primary cultured uterine leiomyoma cells. We classified 44 surgically removed uterine leiomyomas into four groups according to GnRH agonist use and *MED12* mutation status. CDK8 was co-immunoprecipitated from leiomyoma tissue extracts using MED12 antibody to test its kinase activity *in vitro*, and the amount of phosphorylated substrate was measured. Cell proliferation and apoptosis of primary cultured *MED12* wild-type leiomyoma cells were evaluated in the presence of a CDK8 inhibitor and sex steroid hormones. Of the 44 leiomyomas tested, 11 *MED12* wild-type leiomyomas without preoperative GnRH agonist treatment had significantly higher CDK8 activity than nine GnRH agonist-treated *MED12* wild-type leiomyomas and 15 leiomyomas with *MED12* mutations without GnRH agonist treatment. Treatment of primary cultured *MED12* wild-type cells with CDK8 inhibitors significantly inhibited cell growth and increased apoptosis. *MED12* wild-type leiomyoma cells without GnRH agonist treatment showed high CDK8 activity, and inhibition of CDK8 activity suppressed cell growth *in vitro*.

**Data availability statement:** All relevant data are within the paper and its Supporting information files.

**Funding:** This study was supported in part by the Kihara Memorial Yokohama Foundation for the Advancement of Life Sciences (https://kihara.or.jp/) to K.N., the Kanto-Rengo Society of Obstetrics and Gynecology (https://jsog-k.jp/) ,grant number 2021-01 to K.N., and a Grant-in-Aid for Scientific Research from the Japan Society for the Promotion of Science (https://www.jsps.go.jp/j-grantsinaid/index.html), grant number JP19K09828, and JP22K09599 to R.A.. The sponsors did not play any role in the study design, data collection and analysis, decision to publish, or preparation of the manuscript.

**Competing interests:** K.T. is employed by Shin Nippon Biomedical Laboratories, Ltd., Kagoshima, Japan. The remaining authors declare no competing interests. This does not alter our adherence to PLOS One policies on sharing data and materials.

## Introduction

Uterine leiomyomas (leiomyomas) are the most common benign pelvic tumors in women [1]. Their growth is sex steroid hormone-dependent. Gonadotropin-releasing hormone (GnRH) agonists effectively treat leiomyoma but are not well-tolerated for long periods due to low estrogen-associated menopausal symptoms and osteoporosis. Leiomyoma regrowth upon treatment discontinuation represents a major challenge [2,3]. Many patients require surgical removal of the tumor due to a lack of appropriate drug therapy, highlighting the need for non-hormone modulating therapies.

The genetic background of leiomyomas has become increasingly clear. Somatic mutations in exon 2 of the gene encoding mediator complex subunit 12 (*MED12*) are most common, found in approximately 70% of leiomyomas [4,5]. The most frequent chromosomal structural abnormalities include 12q15 rearrangement (approx. 20% of cases), followed by 7q22 deletion and 6p21 rearrangement [6]. *MED12* wild-type (WT) leiomyomas are larger and more often found as solitary nodules compared to *MED12* mutant (MUT) leiomyomas [5,7]. We previously reported high erythropoietin expression in *MED12* WT leiomyomas associated with intra-tumor blood vessel maturation [8,9]. Active leiomyoma cell proliferation has been observed in *MED12* WT leiomyomas, whereas an increase in the extracellular matrix (ECM) is prominent in *MED12* MUT leiomyomas [10]. The presence of *MED12* mutations can be inferred using magnetic resonance imaging [11,12]. Understanding the growth mechanisms of *MED12* WT and MUT leiomyomas may help guide individualized treatment options for patients.

Cyclin-dependent kinases (CDKs) are broadly classified into those that regulate the cell cycle (e.g., CDK4 and CDK6) or transcription (e.g., CDK8 and CDK9). CDK8 forms a complex with MED12, MED13, and cyclin C, modulating transcription by phosphorylating the second and fifth serine residues in the tandem repeat sequence Tyr-Ser-Pro-Thr-Ser-Pro-Ser (YSPTSPS) in the C-terminal domain (CTD) of RNA polymerase II (RNAP II) [13–15]. CDK8 has gained attention as an oncogenic protein [16], positively regulating transcriptional elongation in the serum response network and contributing to multiple tumorigenic phenotypes [15]. CDK8 inhibition has been explored in estrogen receptor (ER)-positive breast cancer [17], acute myeloid leukemia [18], prostate cancer [19], and glioblastoma [20]. In ER-positive breast cancer, CDK8 inhibitor suppressed estradiol-stimulated RNAP II CTD binding to growth regulating estrogen receptor binding 1 (GREB1) and ER-dependent transcription [17]. This prompted us to consider a similar effect of CDK8 inhibition in leiomyomas.

Turunen et al. reported that CDK8 activity is reduced in *MED12* MUT leiomyomas, but the underlying molecular mechanism and its clinical relevance remain unknown [21–23]. Herein, we hypothesized that CDK8 activity contributes to the tumor growth of *MED12* WT leiomyomas and is influenced by sex steroid hormone kinetics. We, therefore, explored CDK8 as a therapeutic target and investigated its activity in leiomyoma in relation to *MED12* status and sex steroid hormone kinetics using clinical samples.

## Materials and methods

### Patients and tissue samples

We analyzed 46 leiomyoma samples and 2 normal myometrium samples obtained from 44 premenopausal women who underwent total hysterectomy or myomectomy between January 4, 2015, and December 27, 2019. Clinical data extracted from medical records were anonymized and securely maintained in a database at the time of sample collection. Between April 1, 2021, and August 31, 2025, the authors accessed the database for the purposes of this study. The authors did not have access to any information that could identify individual participants during or after data collection. Of these 46 samples, 24 had been utilized in our previous studies [9,11]. *MED12* exon 2 hotspot mutation status was determined as previously described [9]. In brief, DNA was extracted and amplified by PCR using the following primers: GCCCTTTCACCTTGTTCCTT (forward) and TGTCCCTATAAGTCTTCCCAACC (reverse). Mutations were identified by Sanger sequencing. The size reduction rate of leiomyomas with GnRH agonist treatment (monthly subcutaneous injection of 3.75 mg leuprorelin acetate for 2–5 months) was calculated as previously described [11].

This study was approved by the Institutional Review Board of Yokohama City University Graduate School of Medicine (No. A120726018), and it followed the ethical standards for human experimentation established in the Declaration of Helsinki. Written informed consent was obtained from all study participants.

### Preparation of primary cultured leiomyoma cells

Four primary cultures of *MED12* WT leiomyomas and two primary cultures of normal myometrium were isolated as previously described [9] and stored at –145°C in Bambanker (Nippon Genetics Co., Ltd., Tokyo, Japan) until use. The expression of α-smooth muscle actin in these cells has been immunohistochemically demonstrated in our previous study [9]. The stored cells were reconstituted in Dulbecco's modified Eagle's medium (DMEM) with 10% fetal bovine serum (FBS) and 1% penicillin/streptomycin solution at 37°C and 5% $CO_2$. Cultured cells were collected by gentle detachment with Accutase (Nacalai Tesque, Inc., Kyoto, Japan), cultured in phenol red-free DMEM supplemented with charcoal-stripped FBS (10%), and replated at appropriate cell numbers for subsequent assays. Cultured cells passaged only once were used.

### CDK8 activity assay

A GST-conjugated RNAPII CTD was prepared for CDK8 activity assays. The cDNA fragment encoding the CTD was subcloned into pGEX-6P-1 (GE Healthcare Technologies Inc., Chicago, IL). The recombinant protein was expressed in *Escherichia coli* strain BL21 (DE3) (Thermo Fisher Scientific, Waltham, MA) and purified using a Glutathione Sepharose 4B column (GE Healthcare) followed by dialysis with a Slide-A-Lyzer dialysis device (Thermo Fisher Scientific).

A CDK8 activity assay for leiomyoma tissues or primary cultured cells was performed based on the method of Turunen et al., with some modifications [21]. Briefly, protein extracts of leiomyoma tissues or cells were prepared in lysis buffer containing 40 mM Tris-HCl, pH 7.5, 500 mM NaCl, 0.5% deoxycholic acid (sodium salt), 1% Triton X-100, and 1 mM EDTA, supplemented with protease and phosphatase inhibitor cocktails (Millipore Sigma, Burlington, MA). Protein concentration was determined using a Micro BCA Protein Assay Kit (Thermo Fisher Scientific) and standardized to 9.3 mg/mL (tissues) or 1.2 mg/mL (cells) in buffer. Rabbit polyclonal anti-MED12 antibody (1 µg) (Bethyl Laboratories, Inc., Montgomery, TX) was cross-linked to 12.5 µL Dynabeads protein G (Thermo Fisher Scientific) and mixed with 380 µL (tissues) or 1 mL (cells) of protein extract at 4°C overnight. Dynabeads crosslinked with non-specific rabbit IgG served as controls. The Dynabeads-immunoprecipitate (IP) was collected using a magnet. CDK8 kinase activity in the precipitate was directly assessed using an *in vitro* kinase reaction with 2 µg of GST-fused CTD protein in 25 µL of kinase buffer (20 mM Tris-HCl, pH 7.9, 20% glycerol, 50 mM KCl, 20 mM $MgCl_2$, 20 µg/mL BSA, and 1 mM dithiothreitol) with 350 µM ATP at 30°C for 10 min with shaking. The reaction mixture was separated into Dynabeads-IP and supernatant using a magnet, heated at 100°C to stop the reaction, and subjected to western blotting.

## Western blotting

For CDK8 kinase activity assays, 10 µL of the reaction supernatant (CTD), Dynabeads-IP, and 12 µL of the extracts before kinase reaction (input) were prepared for western blotting by adding NuPAGE LDS sample buffer (Thermo Fisher Scientific) as per the manufacturer's instructions. For Dynabeads-IP, 25 µL of the sample buffer was added and heated at 90°C for 3 min; subsequently, 15 µL of the liquid fraction was subjected to western blotting.

Each sample was electrophoresed in NuPAGE 4–12% gradient Bis-Tris Protein Gel (Thermo Fisher Scientific), transferred to polyvinylidene difluoride membranes and blotted with anti-RNAPII CTD repeat YSPTpSPS (pS5) (rat monoclonal, 1:1000; Abcam plc, Cambridge, UK) and anti-RNAPII CTD repeat YSPTSPS (rat monoclonal, 1:2000; Abcam) for CTD; anti-CDK8 (rabbit monoclonal, 1:1000, ab229192; Abcam) and anti-MED12 (rabbit monoclonal, 1:1000; Bethyl Laboratories) for Dynabeads IP; and anti-CDK8, anti-MED12, and anti-β-actin (mouse, monoclonal, 1:1000; Millipore Sigma) for input. For the inputs of primary cell culture samples, ER alpha (ERα) was detected using anti-ERα (rabbit, monoclonal, 1:1000; Cell Signaling Technology, Danvers, MA). Corresponding horseradish peroxidase-labeled secondary antibody binding was detected using an enhanced chemiluminescence method (ImmunoStar LD; FUJIFILM, Tokyo, Japan, or Amersham ECL Prime; Cytiva, Marlborough, MA). Images were captured using an ImageQuant LAS4000 digital camera system and analyzed using ImageQuant TL software (GE Healthcare). The intensity of each band was compared with the relative value without correction, and the amount of input protein was corrected using the β-actin value.

## Immunohistochemistry

Formalin-fixed paraffin-embedded tissue sections were deparaffinized, rehydrated, and subjected to heat-induced antigen retrieval in an antigen retrieval solution (pH 9.0) using an autoclave at 110°C for 15 minutes. Endogenous peroxidase activity was blocked by immersion in 3% hydrogen peroxide solution. Subsequent staining was performed using an automated staining device (Histostainer, Nichirei Biosciences Inc., Tokyo, Japan). A ready-to-use anti-human estrogen receptor antibody (#723911, Nichirei Biosciences) was used as the primary antibody. Labeled antigens were detected using Histofine Simplestain MAX-PO (R) and the Histofine DAB Substrate Kit. Nuclei were counterstained with hematoxylin.

ER expression was evaluated using the H-score method, calculated as the sum of the products of the staining intensity (no staining = 0; weak = 1+; moderate = 2+; strong = 3+) and the percentage of stained nuclei. Automated scoring was performed using the Aperio's annotation software (Leica Biosystems, Nussloch, Germany).

## Effect of CDK8 inhibition on cell growth

Primary cultured cells ($1 \times 10^3$ cells/well) were seeded into a 96-well plate, and various concentrations of Senexin A (0, 5, and 10 µM, Cayman Chemical Company, Ann Arbor, MI) or Senexin B (0, 1, 2, and 5 µM, Cayman Chemical Company) were evaluated using the cell counting kit-8 assay (CCK8 assay, Dojindo Laboratories, Kumamoto, Japan) according to the manufacturer's protocol, using a microplate reader (En Spire; Perkin Elmer, Inc., Shelton, CT).

To evaluate cell proliferation after stimulation with sex steroids, primary cultured cells were seeded into a 96-well plate ($1 \times 10^3$ cells/well) and cultured for 24 h. After sex steroid hormone deprivation (24 h), Senexin B (final concentration of 2 µM) or DMSO was added, and the medium was changed every 2 or 3 days with fresh Senexin B-containing medium. To determine the effect of low sex steroid environments by GnRH agonist treatment on CDK8 activity and combination effects with CDK8 inhibitors and low sex steroid status, cells were treated with 10 nM 17β-estradiol (E2) and 100 nM progesterone (P4) (= EP) or left untreated. Cell proliferation was evaluated via the CCK8 assay on days 1, 4, 7, 10, and 13 after steroid addition. The average of six wells was used. All experiments were performed in triplicate.

## CDK8 activity assay in cell culture study

Subsequent experiments were conducted using two types of primary leiomyoma culture cells (LM1 and LM2). For the CDK8 activity assay, $5 \times 10^5$ cells (LM1) or $8 \times 10^5$ cells (LM2) were seeded into 100 mm Petri dishes and incubated for

24 h. Following incubation, steroid deprivation was performed. Proteins were extracted on day 4 after Senexin B and EP treatment, and CDK8 activity was evaluated.

## Quantitative reverse transcription PCR

Primary culture cells ($1 \times 10^5$/well) were plated in a 6-well plate and treated with Senexin B, followed by activation of growth with EP under the same conditions as for the CCK8 assay and cultivation for 72 h. Total RNA was extracted using an RNeasy mini kit (Qiagen N.V., Venlo, Netherlands) and reverse-transcribed using SuperScript VILO Master Mix (Thermo Fisher Scientific) according to the manufacturer's protocol. Quantitative reverse transcription PCR was performed on a Light Cycler 96 System (Roche, Basel, Switzerland) using TaqMan Gene Expression Assay kits (Thermo Fisher Scientific) targeting mRNAs of *ESR1* (Hs01046816_m1), *GREB1* (Hs00536409_m1), and *RNA18S5* (Hs03928985_g1). The cycling conditions were as follows: initial denaturation at 95°C for 1 min, followed by 45 cycles of denaturation at 95°C for 15 s and annealing and extension at 60°C for 60 s. The relative expression levels were analyzed using the comparative $2^{-\Delta\Delta Ct}$ method, with 18S rRNA as the endogenous control.

## Flow cytometry

Cells were seeded into 6-well plates and harvested with Accutase on days 1, 3, 5, and 8 following steroid addition. Cell cycle was assessed using Cell Cycle Assay Solution Deep Red (Dojindo Laboratories). Cells were then subjected to double staining with Annexin V and propidium iodide, and apoptosis was assessed using the Cell Meter™ Annexin V Binding Apoptosis Assay Kit (ATT Bioquest, Pleasanton, CA), on a FACSCanto II equipped with FACS Diva software (BD Biosciences, Franklin Lakes, NJ) using the manufacturer's instructions.

## Statistical analysis

The results of tissue experiments were compared using the Mann–Whitney U test and Kruskal–Wallis test with Dunn–Bonferroni post hoc analysis. All experiments with primary cells were performed three times, and a two-way analysis of variance (Two-way ANOVA) was performed to compare treatment groups for two independent variables and their interaction. If no interaction was detected, the main effect of each drug was assessed. When interaction was present, Tukey's post hoc test was performed to determine specific group differences. Statistical significance was set at $p < 0.05$. Statistical analyses were performed using the IBM SPSS Statistics version 27.0.

## Results

### Clinical characteristics

Of 44 patients, 26 did not receive preoperative GnRH agonist treatment (11 *MED12* WT and 15 *MED12* MUT leiomyomas), and 18 did (9 *MED12* WT and 9 *MED12* MUT leiomyomas). There were no significant differences in age, BMI, number of pregnancies and deliveries, maximum tumor diameter, or multiplicity of leiomyoma growth among the four groups divided based on preoperative GnRH agonist treatment and *MED12* mutation status (Kruskal–Wallis test with Dunn's post hoc analysis). Of the 18 GnRH agonist-treated tumors, *MED12* WT leiomyomas showed a higher shrinkage rate than *MED12* MUT leiomyomas (44.1% vs. 15.6% reduction rate, $p = 0.008$, Mann–Whitney U test) (Table 1).

### CDK8 activity in leiomyoma tissue samples

Levels of CTD phosphorylated at the 5th serine residue (CTD-pS5) were evaluated for assessing CDK8 activity by dividing samples into four groups based on GnRH agonist treatment and *MED12* mutation status (Fig 1A, B).

In the absence of GnRH treatment, *MED12* WT leiomyomas exhibited significantly higher CDK8 activity than *MED12* MUT leiomyomas ($p = 0.02$, Kruskal–Wallis test with Dunn's post-hoc analysis). No difference in CDK8 activity was observed based on *MED12* status in GnRH agonist-treated tumors ($p > 0.99$). In *MED12* WT leiomyomas,

**Table 1. Clinical characteristics.**

| GnRH agonist usage/*MED12* status | GnRHa (–)/ *MED12* WT (n = 11) | GnRHa (–)/ *MED12* MUT (n = 15) | GnRHa (+)/ *MED12* WT (n = 9) | GnRHa (+)/ *MED12* MUT (n = 9) | *p*-value |
|---|---|---|---|---|---|
| Age | 43 (41–47) | 45 (43–45) | 44 (43–48) | 43 (39–44) | 0.23 |
| BMI | 22.6 (21.1–24.9) | 24.9 (22.8–29.9) | 23.9 (21.8–29.7) | 24.9 (21.2–26.1) | 0.38 |
| Gravida | 1 (1–3) | 1 (0–2) | 0 (0–2) | 0 (0–1) | 0.23 |
| Parity | 1 (0–2) | 1 (0–2) | 0 (0–1) | 0 (0–1) | 0.25 |
| Maximum diameter of leiomyoma (cm) | 6.6 (4.7–8) | 7.2 (6.2–8.6) | 8.7 (8–11.4) | 9 (6.9–12.2) | [a]0.03 |
| Multiple leiomyoma | 5 (45.5%) | 13 (86.7%) | 6 (66.7%) | 7 (77.8%) | 0.15 |
| GnRH agonist treatment count (months) | – | – | 3 (3–4.5) | 3 (3–3) | 0.26 |
| Tumor shrinkage rate after GnRH agonist treatment | – | – | 44.1 (22.8–56.6) | 15.6 (11.1–22.0) | 0.008 |

Values are presented as median (interquartile range) or number (%). The Mann–Whitney U test was used for two-group comparisons, and the Kruskal–Wallis test was used for four-group comparisons. The Dunn–Bonferroni test was used for post-hoc analysis. [a]Post hoc analysis showed no significant differences. BMI, body mass index; GnRH, gonadotropin-releasing hormone; MUT, mutant; WT, wild type.

samples not treated with GnRH agonist treatment had significantly higher CDK8 activity than those that were ($p = 0.048$). No such difference was observed for *MED12* MUT leiomyomas ($p > 0.99$) (Fig 1B). Among *MED12* WT samples without GnRH agonist treatment, there were cases with low MED12 expression and CDK8 activity (e.g., Case No. 1), as well as, cases with sufficient input-MED12 but low IP-MED12 and CDK8 activity (e.g., Case No. 3). Input-CDK8 and input-MED12 levels did not differ significantly between the groups, but the amount of CDK8 bound to MED12 (IP-CDK8) and IP-MED12 were significantly greater in *MED12* MUT leiomyomas treated with GnRH agonist (Fig 1C–F). No correlation was found between CDK8 activity and the maximum tumor diameter or shrinkage rate with GnRH agonists ($p = 0.52$ and 0.46, respectively, Spearman rank correlation coefficient; S1 Fig). The H-score from ERα immunohistochemistry was not significant differences between groups, as determined by the Kruskal–Wallis test with Dunn's post hoc analysis. Furthermore, no correlation was observed between ERα expression and CDK8 activity (S2 Fig).

**Effects of CDK8 inhibitor in primary cultured leiomyoma and myometrial cells**

To determine the optimal CDK8 inhibitor concentration, we first tested various concentrations of Senexin A and Senexin B for their growth inhibitory effects in primary cultured leiomyoma cells. Both Senexin A and B exhibited dose-dependent inhibitory effects on cell growth (S3 and S4 Figs). Notably, 2 μM and 5 μM of Senexin B produced similar saturated inhibitory effects on the growth of primary cultured cells (S4 Fig). Based on these findings, we used 2 μM Senexin B for subsequent experiments. We then evaluated cell growth stimulated by EP following sex steroid deprivation. Day 4 onwards, significant growth inhibition by Senexin B was observed in all leiomyoma cell types, while EP promoted cell growth in three out of four cell types (LM1, LM2, and LM3) (Fig 2A). In normal myometrial cells, only the inhibitory effect of Senexin B on cell proliferation was observed; EP had no detectable effect (Fig 2B). EP notably promoted cell growth in several leiomyoma cell types, while Senexin B consistently suppressed proliferation across all leiomyoma and myometrial cells. To assess whether these agents exert a synergistic effect, we examined their combined impact over time. Except in LM3 cells on day 13, no significant interaction between Senexin B and EP was observed, indicating that their effects were predominantly independent rather than synergistic.

The proportion of apoptotic cells in the sub-G1 phase increased after Senexin B treatment (days 5 and 8 of LM1, days 3, 5, and 8 of LM2) (Fig 3A, B). In LM1, the proportion of cells in G1 increased while those in S decreased, whereas in LM2, the proportion of cells in G1 decreased while those in G2 increased (Fig 3C). Annexin V-stained apoptotic cells significantly increased on day 8 of LM1 and days 3 and 5 of LM2 after treatment with Senexin B, and apoptotic increased on the other days. Among Annexin V-positive cells, those negative for propidium iodide staining were interpreted as early

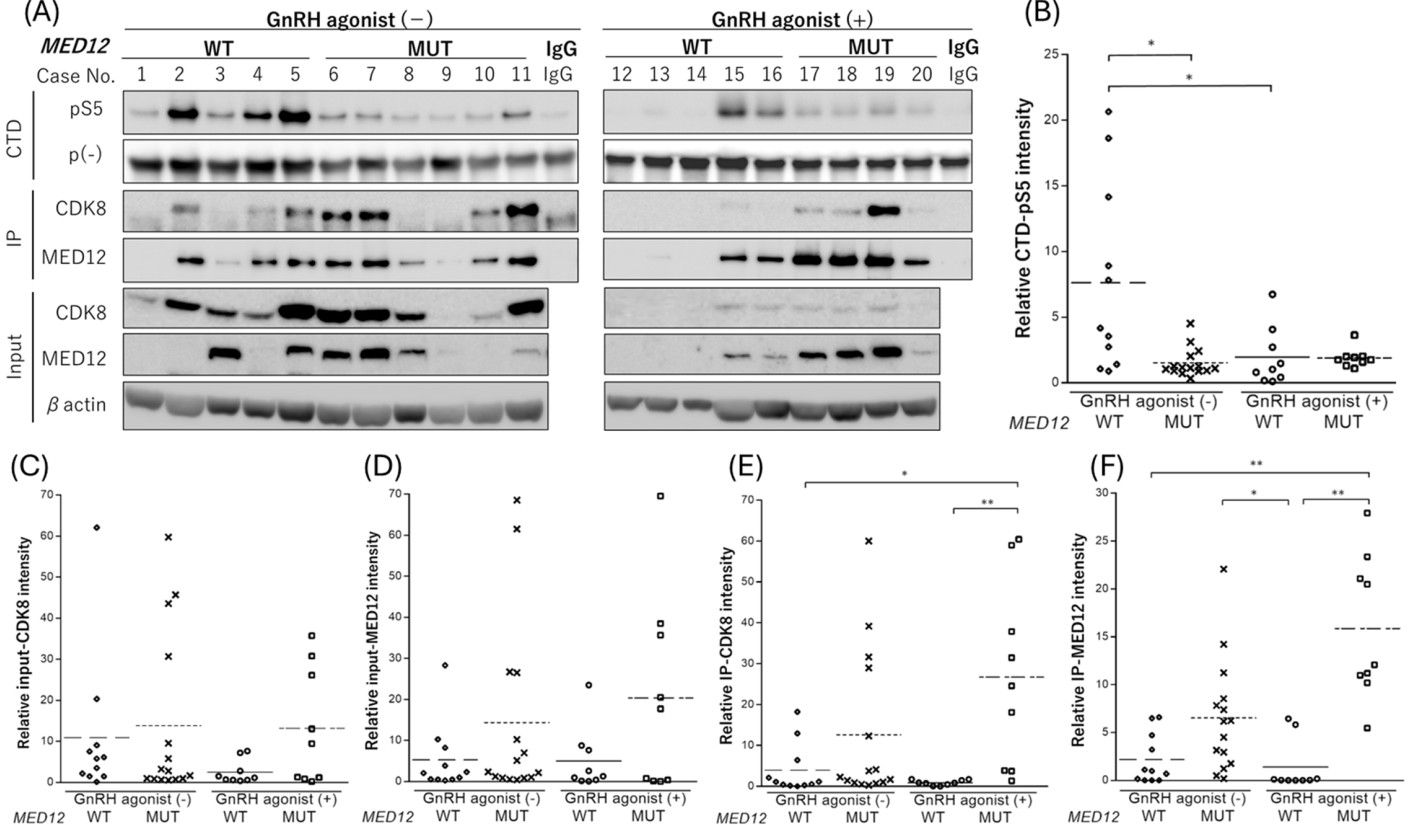

**Fig 1. Western blotting analysis of CDK8 kinase assay in leiomyoma specimens.** (A) Representative western blot image; CTD-pS5 phosphorylated by the CDK8-MED12 complex and non-phosphorylated CTD protein (p–) are indicated in the upper part of the panel. Protein extracts were immunoprecipitated with anti-MED12 antibodies and immunoblotted with anti-CDK8 and anti-MED12 antibodies as indicated in the middle of the panel. Protein extracts were immunoblotted with anti-CDK8, anti-MED12, and anti-β-actin antibodies as indicated in the lower part of the image. (B–F) Graph of each protein band intensity according to *MED12* status (WT or MUT) in GnRH agonist-treated (GnRH agonist+) and -untreated (GnRH agonist–) groups. * $p < 0.05$; ** $p < 0.01$. CTD, C-terminal domain (of the largest subunit of RNA polymerase II); CTD-pS5, C-terminal domain with phosphorylated 5th serine residue; GnRH, gonadotropin-releasing hormone; IP, immunoprecipitated; MUT, mutant; WT, wild type.

apoptosis and those positive were in late apoptosis, the proportion of both of which were increased in LM1, whereas that of early apoptosis cells was increased in LM2 (Fig 4A, B).

Western blotting results are shown in Fig 5A–C. The amount of CTD-pS5 phosphorylated by IP-CDK8 was significantly reduced 4 days after Senexin B addition (LM1: $p = 0.01$, LM2: $p < 0.001$). In contrast, IP-CDK8 levels significantly increased in LM2 and tended to increase in LM1 (LM1: $p = 0.24$, LM2: $p < 0.001$).

ERα protein expression was significantly decreased by Senexin B in LM2 and tended to decrease in LM1 (LM1: $p = 0.20$, LM2: $p < 0.001$). ERα-related mRNA expression was examined, and *ESR1* expression was suppressed by Senexin B treatment (LM1, $p = 0.001$, LM2, $p = 0.002$). Expression of *GREB1*, an estrogen-responsive gene, was increased after EP treatment (LM1, $p = 0.001$, LM2, $p < 0.001$). LM1 exhibited a non-significant decrease in *GREB1* expression under Senexin B treatment ($p = 0.61$), whereas LM2 showed a significant decrease ($p = 0.01$) (Fig 6).

## Discussion

In this study, we demonstrated that MED12-CDK8 complex-mediated RNAPII CTD phosphorylation was higher in *MED12* WT than in MUT leiomyomas and was effectively attenuated by GnRH agonist treatment. In primary cultured *MED12* WT cells, CDK8 inhibitor increased apoptosis and suppressed cell proliferation.

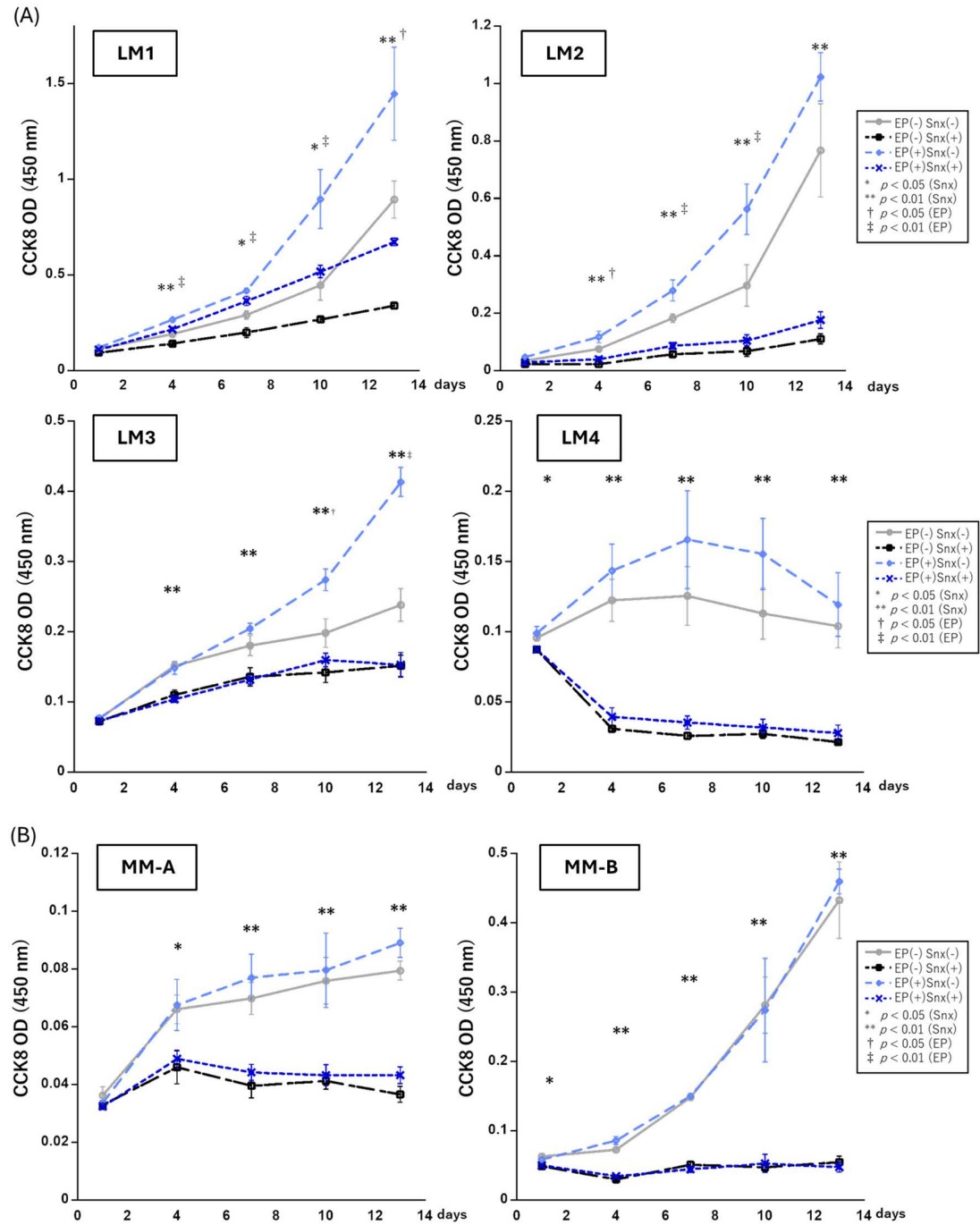

**Fig 2. Effect of Senexin B on the proliferation of primary leiomyoma and myometrial cells.** (A) The time course of optical density values measured via CCK8 assay on days 1, 4, 7, 10, and 13 after the addition of the drug (Snx±EP); leiomyoma cells derived from four patients. (B) CCK8 assay of normal myometrial cells. MM-A and MM-B were derived from the same patient as LM3 and LM4, respectively. Two-way ANOVA was used for the analysis, and there was no interaction between Snx and EP, except on day 13 of LM3. The time point with a significant difference in Snx is denoted (*, $p < 0.05$; **, $p < 0.01$), and the time point with a significant difference in EP is denoted (†, $p < 0.05$, ‡, $p < 0.01$). On day 13 of LM3, an interaction between Snx and EP was observed ($p = 0.002$); thus, Tukey's test was performed. Compared to the EP (+) Snx (–) group, the other groups showed a significant decrease in cell proliferation ($p < 0.001$). CCK8, cell counting kit-8; EP, 17-β estradiol (E2) + progesterone (P4); Snx, Senexin B.

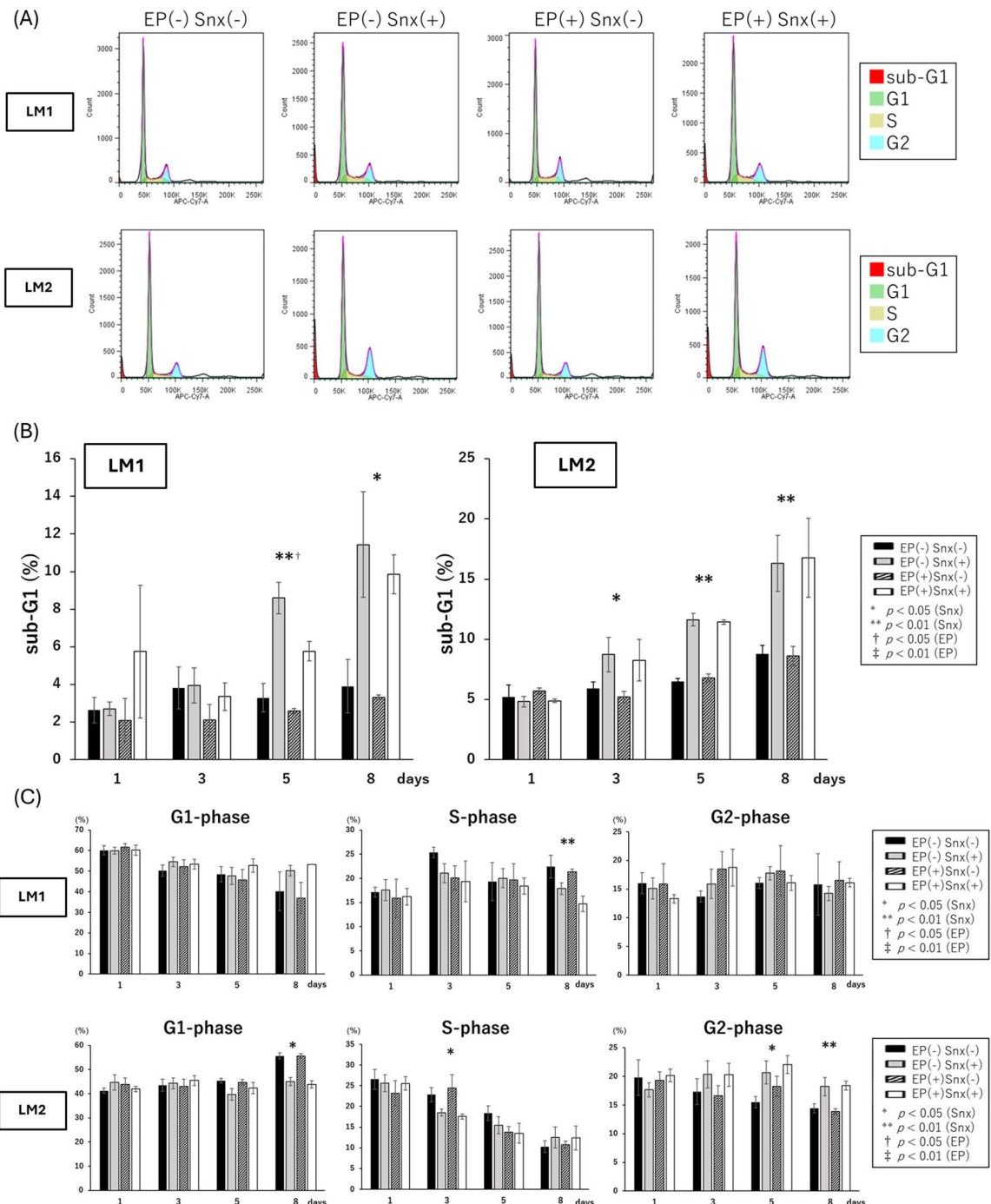

**Fig 3. Effect of Senexin B on the cell cycle.** (A) Representative histogram of the cell cycle measured by flow cytometry. Primary cultured cells treated with Snx±EP on day 8 are shown. The red area indicates the sub-G1 phase, representing apoptotic cells. (B, C) The percentage of cells in the sub-G1, G1, S, and G2 phases on days 1, 3, 5, and 8. Two-way ANOVA showed no interaction between Snx and EP. The time point with a significant difference in Snx is denoted (*, $p < 0.05$; **, $p < 0.01$), and the time point with a significant difference in EP is denoted (†, $p < 0.05$, ‡, $p < 0.01$). EP, 17-β estradiol (E2) + progesterone (P4); Snx, Senexin B.

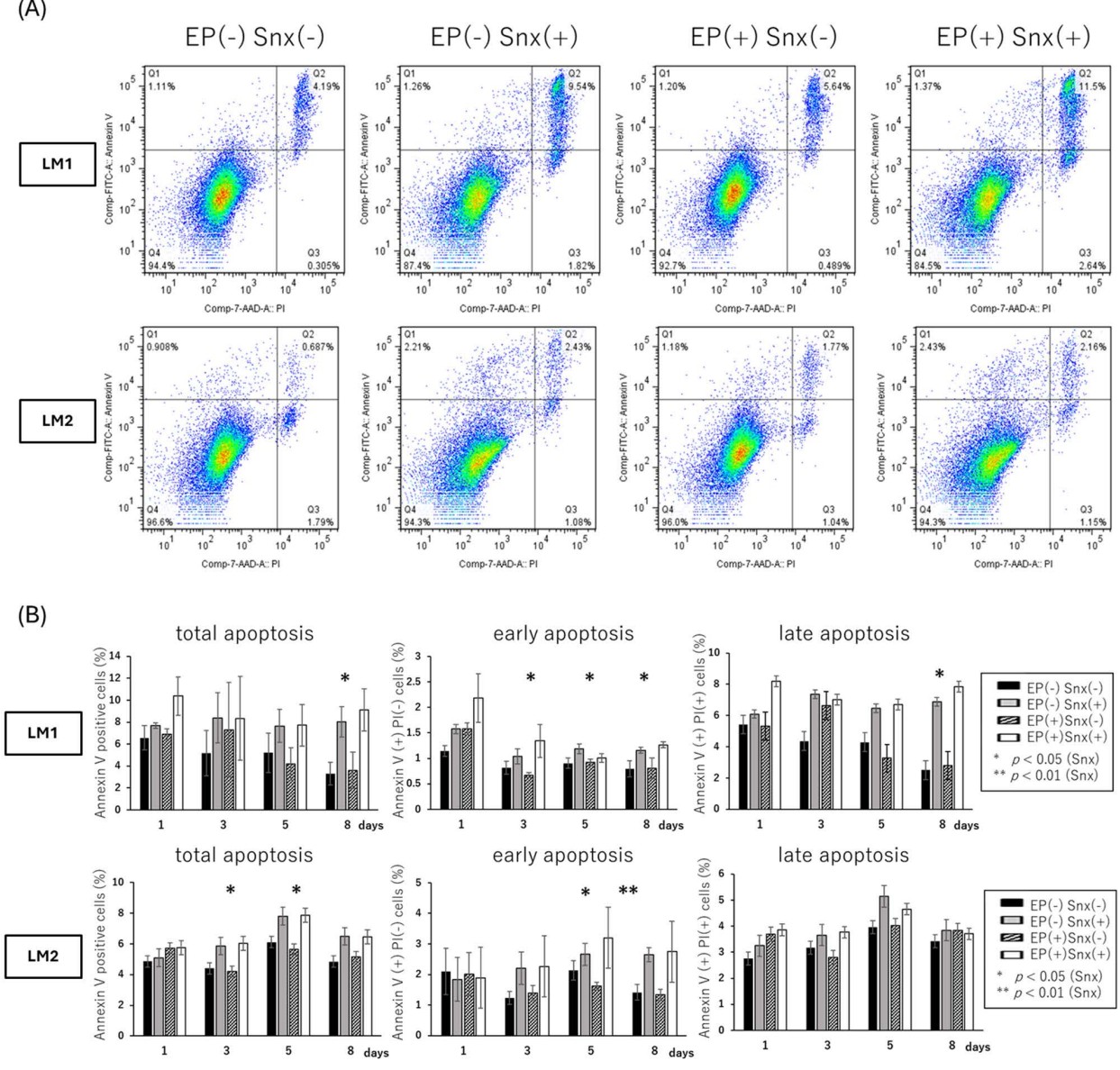

**Fig 4. Flow cytometric analysis of apoptosis using Annexin V/PI staining.** (A) Representative flow cytometry dot plot of Annexin V/PI double staining on day 8. (B) Time-course analysis of total apoptosis (Annexin V-positive cells), early apoptosis (Annexin V-positive/PI-negative), and late apoptosis (Annexin V-positive/PI-positive). Two-way ANOVA showed no interaction between Snx and EP. The time point with a significant difference in Snx is denoted (*, p < 0.05; **, p < 0.01). EP, 17-β estradiol (E2) + progesterone (P4); PI, propidium iodide; Snx, Senexin B.

Although *MED12* exon 2 mutations and CDK8 activity have been reported in leiomyoma cells and tissues [21–23], this is the first report clarifying their association with patient treatment history. Turunen et al. reported that *MED12* mutations inhibited its binding to cyclin C and reduced CDK8 activity [21]. Park et al. later showed that *MED12* mutations did not inhibit cyclin C binding, yet CDK8 kinase activity was reduced [22,23]. Our results reinforced previous findings by showing significantly reduced CDK8 kinase activity towards RNAPII CTD in *MED12* MUT leiomyomas. This reduction occurred even though normal levels of MED12 and CDK8 proteins were co-immunoprecipitated using an anti-MED12 antibody.

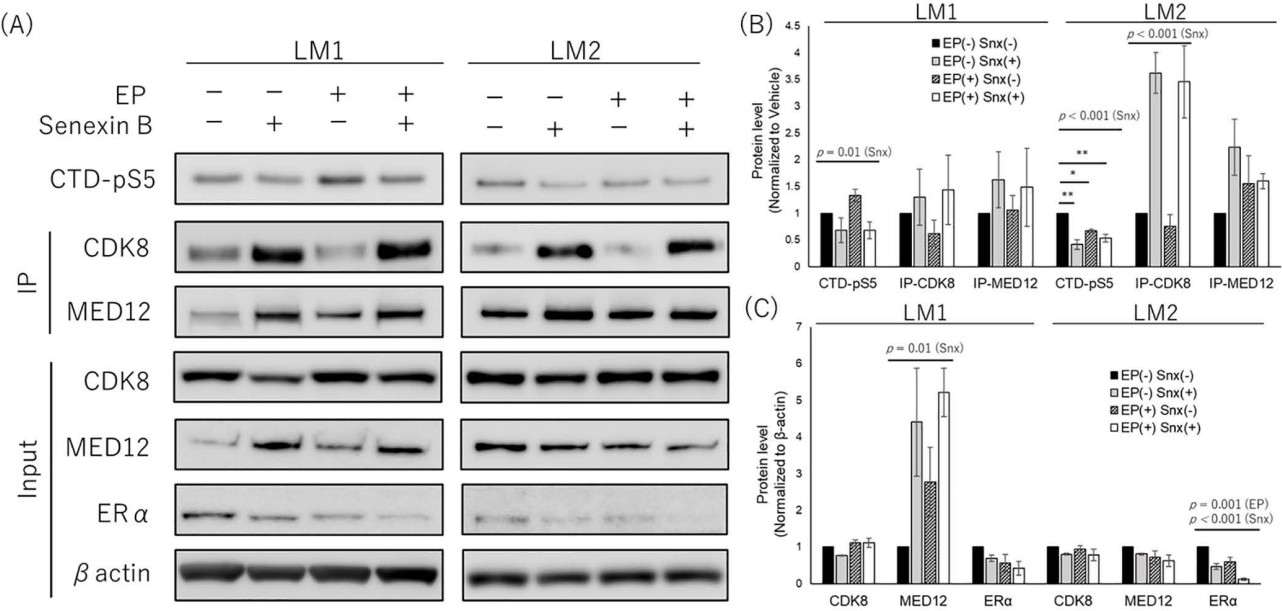

**Fig 5. Western blotting analysis in primary cells treated with Senexin B.** Primary leiomyoma cells were treated with Snx ± EP for 4 days followed by CDK8 phosphorylation assay and western blotting analysis. Results for the leiomyoma cells derived from two patients (LM1 and LM2) are shown. Data are normalized to vehicles and presented as the means of three independent experiments. (A) Representative western blot image: the CTD-pS5 protein is indicated in the upper part of the panel. Cell extracts were immunoprecipitated with an anti-MED12 antibody and immunoblotted with anti-CDK8 and anti-MED12 antibodies, as indicated in the middle of the panel. Protein extracts were immunoblotted with anti-CDK8, anti-MED12, and anti-ERα antibodies, as indicated in the lower part of the panel. Anti-β-actin was used as a loading control. (B) Quantification of phosphorylated CTD protein and immunoprecipitated protein. (C) Quantification of protein extracts. Protein levels were quantified, and expression data are presented relative to β-actin. The data are normalized to the control and presented as the mean of three independent experiments. Two-way ANOVA showed an interaction between Snx and EP at CTD-pS5 in LM2, and post-hoc test showed a significant difference between EP (–) Snx (–) and other groups. (*, $p < 0.05$; **, $p < 0.01$). CTD, C-terminal domain (of the largest subunit of RNA polymerase II); CTD-pS5, C-terminal domain with phosphorylated 5th serine residue; EP, 17-β estradiol (E2) + progesterone (P4); ERα, estrogen receptor alpha; IP, immunoprecipitated; Snx, Senexin B.

Furthermore, CDK8 inhibition in primary cultured *MED12* WT cells increased the amount of CDK8 immunoprecipitated with anti-MED12 antibody (IP-CDK8) and decreased CDK8 kinase activity (CTD-pS5), as also observed in *MED12* MUT tissues.

CDK8 inhibitors have been shown to suppress proliferation in several types of cancer, including ER-positive breast cancer and acute myeloid leukemia [17–20]. However, heterogeneous responses have been reported; for instance, in prostate cancer cell lines, the response has been variable [19], while in colon cancer, only long-term treatment showed growth inhibition [24]. Here, we found that Senexin B inhibited cell growth and increased apoptosis in *MED12* WT primary cultured cells, as evidenced by sub-G1 accumulation and Annexin V staining. Nakamura et al. reported that CDK8 inhibitors increase the sub-G1 fraction by regulating the G1/S transition [19]. Consistent with these findings, our experiments showed increased sub-G1 fractions. However, the accompanying changes in other cell-cycle phases (G1, S, and G2) varied across cell types, suggesting that the impact on cell-cycle distribution is not uniform but rather cell type dependent. Since CDK8 inhibition primarily drives transcriptional reprogramming rather than canonical cell cycle control [17,25,26], the net impact on phase distribution is likely cell intrinsic and varies across cell types.

Although the mechanism by which decreased CDK8 activity suppresses leiomyoma growth remains unclear, this study revealed a GnRH agonist-induced reduction in MED12-dependent CDK8 activity. Compared to MUT tumors, *MED12* WT leiomyomas have previously shown greater shrinkage with GnRH agonist treatment [11]. In the current study, *MED12* WT tissues treated with GnRH agonists exhibited both stronger tumor shrinkage and a lower level of CDK8 activity compared

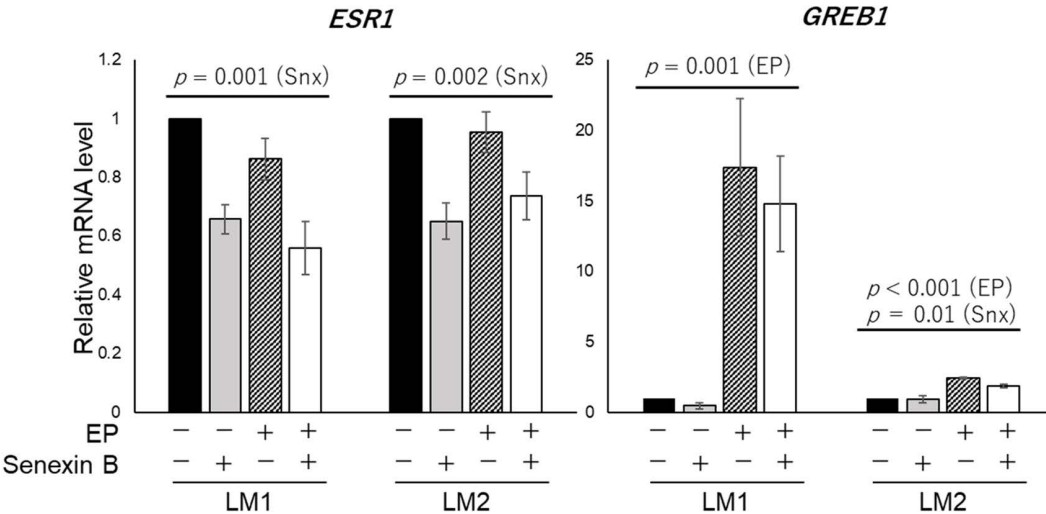

**Fig 6. Changes in mRNA expression in primary cells treated with Senexin B.** Primary leiomyoma cells were treated with Snx±EP for 72 h, RNA was extracted, and qRT-PCR was performed. Two-way ANOVA showed no interaction between Snx and EP. EP, 17-β estradiol (E2) + progesterone (P4); Snx, Senexin B.

to the untreated group. Conversely, *MED12* MUT tissues showed less shrinkage and exhibited low CDK8 activity regardless of GnRH agonist treatment. This difference in baseline CDK8 activity between WT and MUT tumors may contribute to the observed differential efficacy of GnRH agonists. However, this relationship remains unconfirmed because pretreatment CDK8 activity was not assessed. To test whether low estrogen–progesterone (EP) conditions mimic GnRH agonist treatment *in vitro*, we compared CDK8 activity in cultured cells with and without EP supplementation. CDK8 activity did not change in the presence or absence of EP, indicating that low EP alone does not affect CDK8 activity. The effect of other known mechanisms of GnRH agonist therapy, such as reduced blood flow and degenerative changes in leiomyoma tissue [27,28], should be further evaluated. The role of GnRH agonists in apoptosis remains controversial, with several studies reporting no significant effects [29,30]. In our study, apoptosis was not induced under EP-deprived conditions but was observed with Senexin B treatment. These results suggest that CDK8 inhibition could be effective in patients who do not respond to GnRH agonist therapy.

A study of ER-positive breast cancer reported that MED12 knockdown reduced ERα protein and mRNA expression [31]. Studies on CDK8 inhibitors for breast cancer have reported no change in ER expression, but prevented induction of ER-responsive genes by estrogen addition to estrogen-depleted cells. They showed CDK8 is recruited, along with ER, to the *GREB1* promoter upon estrogen stimulation [17]. Herein, CDK8 inhibition in *MED12* WT primary cultured cells tended to decrease ERα protein, *ESR1* and *GREB1* expression. Thus, the MED12–CDK8 complex in leiomyomas appears to be involved in ER transcription. Whether ERα expression in tissue is influenced by *MED12* status or CDK8 activity remains elusive. In the present study, the menstrual cycle phase was not considered at the time of tissue collection and may have influenced our results in clinical samples, because lower ERα expression is typically observed during the secretory phase [32]. Additionally, CDK8 inhibitors suppressed cell proliferation even in the absence of E2, indicating that their effect is not solely attributable to reduced ERα expression or decreased E2 signaling. Although ERα expression may increase under low sex steroid conditions, the contribution of this change to the antiproliferative effect of CDK8 inhibition remains uncertain and should be clarified in future studies.

CDK8 inhibitors are recently developed anti-cancer drugs with the potential to inhibit leiomyoma growth. Although they are teratogenic [33,34], the toxicity in adults is considered low, suggesting minimal side effects [35]. Our cell proliferation

assays demonstrated that Senexin B also inhibits proliferation in myometrial cells. Although this effect is not specific to leiomyomas, suppression of proliferation throughout the uterus may be therapeutically advantageous. *MED12* MUT leiomyomas have been reported to be highly responsive to ulipristal acetate treatment [36], and if CDK8 inhibitors prove effective for *MED12* WT leiomyomas, they may facilitate precision medicine approaches. In addition to tumor growth inhibition, CDK8 inhibitors have also been shown to suppress osteoclast function [37,38], suggesting their potential as prophylactic and therapeutic agents for osteoporosis. The most undesirable effect of long-term GnRH agonist use is osteoporosis caused by low estrogen levels. Therefore, CDK8 inhibitors could offer significant benefits in managing both leiomyoma growth and osteoporosis. However, they are still in early clinical trials, and their safety remains to be established.

This study has several limitations. There were some *MED12* WT cases with low MED12-dependent CDK8 activity, reflecting heterogeneity. Leiomyomas have individual differences in their components, including the level of degeneration, cell density, and percentage of ECM. Because we corrected for the concentration of total extracted protein, it is possible that leiomyomas with a low percentage of nuclei had lower CDK8 activity, but we did not exclude cases because this reflects the true CDK8 activity of the leiomyoma. The association between genomic abnormalities in *MED12* WT and CDK8 activity needs to be examined in a larger cohort.

In leiomyomas, both smooth muscle cells and the ECM contribute to tumor growth and should be considered. In this study, we did not evaluate ECM-related changes. However, since ECM dynamics are strongly influenced by the tumor microenvironment, future *in vivo* studies are essential to clarify their role. In addition, although our analysis focused on ER expression in relation to CDK8 activity, the progesterone receptor (PR) also plays a key role in leiomyoma growth [39]. Previous studies have shown that PR signaling is particularly activated in *MED12* MUT leiomyomas [36,40] and CDK8 inhibition may influence PR expression or activity in *MED12* WT leiomyomas. Moreover, the effects of CDK8 inhibition on *MED12* MUT cells were not assessed due to the difficulty in culturing them [41]. We aim to address these limitations and evaluate the efficacy of CDK8 inhibitors in a mouse leiomyoma xenograft model in future studies.

## Conclusions

In *MED12* wild-type leiomyoma, GnRH agonist treatment decreased CDK8 activity, and CDK8 inhibition suppressed cell proliferation while promoting apoptosis. Therefore, CDK8 inhibitors may also be available for the treatment of leiomyomas.

## Supporting information

**S1 Fig. Relationship between the amount of CTD-pS5 and clinical background.** Spearman's rank correlation coefficient was used for the analysis. (A) Correlation analysis between the amount of CTD-pS5 and maximum tumor diameter of the leiomyoma. (B) Correlation analysis between the rate of leiomyoma size reduction before and after GnRH agonist use and the amount of CTD-pS5. CTD-pS5, C-terminal domain with phosphorylated 5th serine residue; GnRH, gonadotropin-releasing hormone.
(TIF)

**S2 Fig. Examination of ERα expression in leiomyoma tissues.** The H-score for ERα in immunostaining was calculated. (A) Comparison of ERα expression between *MED12* status (WT or MUT) in GnRH agonist-treated (GnRH agonist+) and -untreated (GnRH agonist–) groups. (B) Correlation analysis between ERα expression and the amount of CTD-pS5. Spearman's rank correlation coefficient was used for the analysis. CTD-pS5, C-terminal domain with phosphorylated 5th serine residue; ERα, estrogen receptor alpha; GnRH, gonadotropin-releasing hormone. MUT, mutant; WT, wild type.
(TIF)

**S3 Fig. Effect of Senexin A on the proliferation of primary leiomyoma cells.** Results of CCK8 cell proliferation assay when Senexin A was added at 5 and 10 μM.
(TIF)

**S4 Fig. Concentration-dependent cellular suppression of Senexin B.** Cell proliferation was measured by the CCK8 assay by adding various concentrations of Senexin B to primary cultured leiomyoma cells. Two-way ANOVA revealed significant effects of dose, time, and their interaction ($p < 0.001$ for all). Post hoc analyses using Tukey's test showed that 2 µM and 5 µM significantly inhibited cell growth compared with 1 µM ($p = 0.001$, $p < 0.001$), while no further suppression was observed between 2 and 5 µM ($p = 0.76$).
(TIF)

**S1 Raw Images. Raw western blot images for CDK8 activity assays of leiomyoma tissues.**
(ZIP)

**S2 Raw Images. Raw western blot images for CDK8 activity assays of primary cultured cells.**
(ZIP)

**S1 Dataset. All experimental data supporting this study** .
(XLSX)

## Acknowledgments

We thank Dr. Iwaizumi, Ms. Inada, Ms. Takahashi, and Ms. Komori for their contributions to the experimental work; Ms. Yoshihara for providing support for Sanger sequencing technology; Mr. Nakamura for his assistance with immunohisto-chemistry; and Dr. Kouro for his expertise in flow cytometry.

## Author contributions

**Conceptualization:** Ryoko Asano.

**Funding acquisition:** Koichi Nagai.

**Investigation:** Saki Tanioka.

**Methodology:** Yukihide Ota, Yohei Miyagi.

**Resources:** Koichi Nagai, Katsuya Takenaka.

**Supervision:** Etsuko Miyagi.

**Writing – original draft:** Saki Tanioka.

**Writing – review & editing:** Ryoko Asano, Taichi Mizushima, Yohei Miyagi.

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
