## [Decision Letter · Decision Letter 0]

20 Jul 2025

*MED12*

Dear Dr. Asano,

Thank you for submitting your manuscript to PLOS ONE. After careful consideration, we feel that it has merit but does not fully meet PLOS ONE’s publication criteria as it currently stands. Therefore, we invite you to submit a revised version of the manuscript that addresses the points raised during the review process.

We look forward to receiving your revised manuscript.

Kind regards,

Kazunori Nagasaka

Academic Editor

PLOS ONE

Journal Requirements:

“I have read the journal's policy and the authors of this manuscript have the following competing interests: K.T. is employed by the Shin Nippon Biomedical Laboratories, Ltd., Kagoshima, Japan. The remaining authors report no conflict of interest related to the subject matter of the manuscript.”

We note that one or more of the authors are employed by a commercial company: Shin Nippon Biomedical Laboratories, Ltd., Kagoshima, Japan

3. Please note that your Data Availability Statement is currently missing the repository name and/or the DOI/accession number of each dataset OR a direct link to access each database. If your manuscript is accepted for publication, you will be asked to provide these details on a very short timeline. We therefore suggest that you provide this information now, though we will not hold up the peer review process if you are unable.

4. Please include captions for your Supporting Information files at the end of your manuscript, and update any in-text citations to match accordingly. Please see our Supporting Information guidelines for more information: http://journals.plos.org/plosone/s/supporting-information .

Additional Editor Comments:

Dear Authors,

After careful review, our decision is "Major Revision."

Please address each comment raised by the reviewers carefully and provide a detailed, point-by-point rebuttal letter along with your revised manuscript.

We look forward to receiving your revised manuscript soon.

Sincerely,

Kazunori Nagasaka

Reviewers' comments:

Reviewer's Responses to Questions

**Comments to the Author**

1. Is the manuscript technically sound, and do the data support the conclusions?

Reviewer #1: Yes

Reviewer #2: Yes

Reviewer #3: Yes

2. Has the statistical analysis been performed appropriately and rigorously?

Reviewer #1: No

Reviewer #2: Yes

Reviewer #3: Yes

3. Have the authors made all data underlying the findings in their manuscript fully available?

Reviewer #1: Yes

Reviewer #2: Yes

Reviewer #3: Yes

4. Is the manuscript presented in an intelligible fashion and written in standard English?

Reviewer #1: Yes

Reviewer #2: Yes

Reviewer #3: Yes

Reviewer #1: Manuscript Title:

Impact of MED12 mutation and CDK8 activity on uterine leiomyoma growth and response to gonadotropin-releasing hormone agonist treatment

This study investigates the relationship between MED12 mutation status, CDK8 kinase activity, and the response to GnRH agonist treatment in uterine leiomyomas. The authors analyzed 44 leiomyoma samples classified by MED12 mutation status and GnRH agonist use, and measured CDK8 activity using immunoprecipitation-based kinase assays. In addition, the effects of CDK8 inhibition (Senexin B) were evaluated in primary cultured MED12 wild-type leiomyoma cells.

The study’s key findings are as follows:

1. CDK8 activity is elevated in MED12 wild-type leiomyomas without GnRH agonist treatment.

2. CDK8 activity is suppressed by GnRH agonist treatment.

3. CDK8 inhibition leads to reduced cell growth, increased apoptosis, and decreased expression of estrogen signaling markers (ERα and GREB1) in vitro.

The authors propose that CDK8 activity is hormone-sensitive and that CDK8 inhibitors may provide a novel therapeutic strategy for MED12 wild-type leiomyomas.

The manuscript is clearly written and well-structured. However, significant revisions are necessary to strengthen the mechanistic interpretation and provide further experimental support.

Major Issues

1. Limited in vitro validation of CDK8 inhibition:

The main conclusion—that CDK8 activity promotes leiomyoma cell growth, and its inhibition suppresses proliferation—is currently supported by in vitro data from only two primary cultures. This severely limits the generalizability of the findings. The inclusion of at least one additional primary MED12 WT culture (ideally more than three) is strongly recommended. Furthermore, the authors should evaluate the effect of CDK8 inhibitors on normal myometrial smooth muscle cells to demonstrate leiomyoma-specific vulnerability.

2. Inadequate exploration of the CDK8–ER/PR–hormone axis:

The relationship between CDK8, estrogen/progesterone signaling, and GnRH agonist treatment requires more detailed mechanistic interpretation. If CDK8 inhibition leads to reduced ERα expression, which in turn suppresses estrogen signaling and cell proliferation, this mechanism may not directly explain the tumor shrinkage seen with GnRH agonist treatment, where systemic estrogen levels are already low. Moreover, in vitro, CDK8 inhibition reduced proliferation even without estrogen (E2) supplementation, suggesting that reduced ERα expression alone may not fully account for the observed growth suppression. A more nuanced discussion of these findings is warranted.

Minor Issues

3. The authors should discuss whether the difference in tumor shrinkage between MED12 WT and MUT groups upon GnRH agonist treatment is due to differing baseline CDK8 activity, and whether this explains the differential response.

4. Please clarify how the authors confirmed that the cultured cells were leiomyoma cells rather than fibroblasts (e.g., α-SMA staining).

5. How was the optimal concentration specifically determined in uterine leiomyoma cells? Did the authors assess dose-dependent manner to determine the effective concentration range? This information is essential to ensure the relevance and accuracy of the in vitro findings. It is also recommended to test additional CDK8 inhibitors for reproducibility.

6. Sub-G1 peak analysis alone is insufficient to conclude apoptosis. Please include additional markers such as Annexin-V, cleaved PARP, or Bcl-2 via Western blotting.

7. Reduced cell viability may also result from cell cycle arrest (e.g., G1, G2, or S phase depletion). Please evaluate this possibility.

8. The rationale and biological relevance of adding estradiol and progesterone (EP) in the in vitro culture system should be clearly explained, especially since CDK8 inhibition showed similar effects regardless of EP treatment.

9. Leiomyoma growth is also driven by extracellular matrix (ECM) production. Did CDK8 inhibition alter ECM-related gene or protein expression?

10. Progesterone receptor (PR) plays a key role in leiomyoma cell survival. Was PR expression examined?

11. In Figures 2, 3, and 4, only two-way ANOVA is reported. Post hoc analyses are necessary to clarify which specific group comparisons are statistically significant.

Recommendation: Major Revision

This is a valuable study with potential clinical implications. However, substantial revisions are required to address experimental limitations and refine the mechanistic interpretation.

Reviewer #2: The authors evaluated the impact of MED12 mutation and CDK8 activity on uterine leiomyoma growth and response to Gn-RH agonist treatment. The results are interesting and informative. The manuscript was well written. There were no points to be revised.

Reviewer #3: This manuscript reports on MED12 mutation and CDK8 activity in relation to uterine leiomyoma growth. Overall, the manuscript is well written and the findings are potentially important. However, several essential details and interpretations are missing. To improve the manuscript for publication, I have the following suggestions:

Comments:

1.　The authors describe the relationship between ERα expression and CDK8 activity in primary cultured cells. However, there is no investigation of steroid receptor expression in surgical specimens. To better clarify the impact of altered CDK8 activity on estrogen responsiveness, I recommend including data, if available, on the expression of ERα or estrogen-responsive genes such as GREB1 in the surgical samples.

2.　In the cell culture experiments, both estradiol (E2) and progesterone (P4) were used. The rationale for adding P4, in addition to E2, should be clearly explained. Moreover, since the authors discuss the involvement of the estrogen receptor in regulating cell proliferation, they should clarify how the potential proliferative effects of P4 were excluded in their interpretation.

3.　P14、L.304–306, “CDK8 inhibition in primary cultured MED12 WT cells increased the amount of CDK8 immunoprecipitated with anti-MED12 antibody and decreased CDK8 kinase activity." However, there are no experimental data on CDK8 kinase activity in the primary cultured cells to support this conclusion. The authors should either provide the supporting data or revise the sentence accordingly.

**Do you want your identity to be public for this peer review?** For information about this choice, including consent withdrawal, please see our Privacy Policy

Reviewer #1: No

Reviewer #2: No

Reviewer #3: No

---

## [Author Response · Author response to Decision Letter 1]

8 Oct 2025

We thank the Academic Editor and the reviewers for their thoughtful comments and suggestions, which have significantly improved the clarity and quality of our manuscript.

Reviewer #1: This study investigates the relationship between MED12 mutation status, CDK8 kinase activity, and the response to GnRH agonist treatment in uterine leiomyomas. The authors analyzed 44 leiomyoma samples classified by MED12 mutation status and GnRH agonist use, and measured CDK8 activity using immunoprecipitation-based kinase assays. In addition, the effects of CDK8 inhibition (Senexin B) were evaluated in primary cultured MED12 wild-type leiomyoma cells.

The study’s key findings are as follows:

1. CDK8 activity is elevated in MED12 wild-type leiomyomas without GnRH agonist treatment.

2. CDK8 activity is suppressed by GnRH agonist treatment.

3. CDK8 inhibition leads to reduced cell growth, increased apoptosis, and decreased expression of estrogen signaling markers (ERα and GREB1) in vitro.

The authors propose that CDK8 activity is hormone-sensitive and that CDK8 inhibitors may provide a novel therapeutic strategy for MED12 wild-type leiomyomas.

The manuscript is clearly written and well-structured. However, significant revisions are necessary to strengthen the mechanistic interpretation and provide further experimental support.

Author Response: Thank you for the reviewer’s summary and thoughtful evaluation. Our detailed responses to each point follow below.

Major Issue 1: Limited in vitro validation of CDK8 inhibition

Reviewer Comment: The main conclusion—that CDK8 activity promotes leiomyoma cell growth, and its inhibition suppresses proliferation—is currently supported by in vitro data from only two primary cultures. This severely limits the generalizability of the findings. The inclusion of at least one additional primary MED12 WT culture (ideally more than three) is strongly recommended. Furthermore, the authors should evaluate the effect of CDK8 inhibitors on normal myometrial smooth muscle cells to demonstrate leiomyoma-specific vulnerability.

Author Response: As suggested by the reviewer, we performed additional experiments using primary MED12 WT leiomyoma cultures (LM3 and LM4) and normal myometrial cells (MM-A and MM-B). In LM3 and LM4, Senexin B treatment significantly suppressed proliferation, consistent with the findings in LM1 and LM2 (new data is shown in Fig. 2A). Similarly, in normal myometrial cells, Senexin B suppressed proliferation (new data is shown in Fig. 2B), suggesting that CDK8 inhibition affects both leiomyoma and normal myometrial cells. In this context, suppressing proliferation in both leiomyoma and myometrium could be acceptable, as it may contribute to overall uterine volume reduction. We have discussed this in Discussion (page 20, lines 432–435).

Major Issue 2: Inadequate exploration of the CDK8–ER/PR–hormone axis

Reviewer Comment: The relationship between CDK8, estrogen/progesterone signaling, and GnRH agonist treatment requires more detailed mechanistic interpretation. If CDK8 inhibition leads to reduced ERα expression, which in turn suppresses estrogen signaling and cell proliferation, this mechanism may not directly explain the tumor shrinkage seen with GnRH agonist treatment, where systemic estrogen levels are already low. Moreover, in vitro, CDK8 inhibition reduced proliferation even without estrogen (E2) supplementation, suggesting that reduced ERα expression alone may not fully account for the observed growth suppression. A more nuanced discussion of these findings is warranted.

Author Response: We thank the reviewer for this important comment. Notably, this study does not provide information that can directly discuss the relation between the decrease in CDK8 activity observed with GnRH agonist-treated tumors and the reduction in ERα expression induced by CDK8 inhibition in vitro. In our study, E2 supplementation did not increase CDK8 activity in vitro. As correctly noted by the reviewer, CDK8 inhibition suppressed proliferation even in the absence of E2. Therefore, we have revised the discussion to acknowledge and discuss that unknown additional mechanisms are likely involved in the effects of GnRH agonists (page 19, lines 405–411; page 20, lines 425–429).

Minor Issues:

1. The authors should discuss whether the difference in tumor shrinkage between MED12 WT and MUT groups upon GnRH agonist treatment is due to differing baseline CDK8 activity, and whether this explains the differential response.

Author Response: MED12 MUT leiomyomas had lower baseline CDK8 activity than WT tumors and showed no further decrease after GnRH agonist treatment. This difference in baseline activity may partly explain the smaller shrinkage observed in MUT compared to WT groups. However, because this relationship could not be directly evaluated in vitro, we described the effect of GnRH agonists on leiomyomas in relation to CDK8 activity as uncertain in the revised manuscript (page 19, lines 398–405).

2. Please clarify how the authors confirmed that the cultured cells were leiomyoma cells rather than fibroblasts (e.g., α-SMA staining).

Author Response: The identity of these cells as leiomyoma cells was confirmed by immunohistochemical demonstration of α-smooth muscle actin expression in our previous study. We have cited the paper which reports these results (page 5, lines 103–104).

3. How was the optimal concentration specifically determined in uterine leiomyoma cells? Did the authors assess dose-dependent manner to determine the effective concentration range? This information is essential to ensure the relevance and accuracy of the in vitro findings. It is also recommended to test additional CDK8 inhibitors for reproducibility.

Author Response: We determined the optimal Senexin B concentration by assessing the dose–response relationship on cell growth. We tested concentrations of 0, 1, 2, and 5 µM, and 2 and 5 µM of Senexin B showed similar saturated growth inhibitory effect on primary cultured cells (S4 Fig). Therefore, we used 2 µM for subsequent experiments. To address the reviewer’s suggestion, we also tested an additional CDK8 inhibitor, Senexin A. As shown in S3 Fig, 5 and 10 µM of Senexin A showed concentration-dependent inhibition of cell proliferation. These points are described in the Materials and Methods (page 8, lines 171–175), and the Results section (page 14, lines 274–278).

4. Sub-G1 peak analysis alone is insufficient to conclude apoptosis. Please include additional markers such as Annexin-V, cleaved PARP, or Bcl-2 via Western blotting.

Author Response: Thank you for highlighting this. We additionally evaluated Annexin V-positive cells using flow cytometry (new data is shown in Fig. 4). We have described double staining for Annexin-V and PI, confirming apoptosis, in Materials and Methods (page 10, lines 205–210) and Results (page 15, lines 307–312).

5. Reduced cell viability may also result from cell cycle arrest (e.g., G1, G2, or S phase depletion). Please evaluate this possibility.

Author Response: A graph showing changes in the G1, S, and G2 phases has been added. LM1 showed a decrease in the proportion of cells in G1 and an increase in the proportion of cells in S, while LM2 showed the opposite. No consistent trend was observed (new data is shown in Fig. 3). We have added a discussion of cell cycle arrest (pages 18–19, lines 389–395).

6. The rationale and biological relevance of adding estradiol and progesterone (EP) in the in vitro culture system should be clearly explained, especially since CDK8 inhibition showed similar effects regardless of EP treatment.

Author Response: To reproduce the low EP environment caused by GnRH agonist treatment, EP (+/–) conditions were set and compared in vitro. We added EP to the cell proliferation assay to examine the combined effects of CDK8 inhibitors and GnRH agonists, but no synergistic effects were evident. CDK8 activity was not altered by the addition of EP, suggesting that the low EP state of GnRH agonists does not reduce CDK8 activity. Although negative, these findings are important for understanding the hormonal context of CDK8 activity (page 19, lines 405–414; page 20, lines 425–429).

7. Leiomyoma growth is also driven by extracellular matrix (ECM) production. Did CDK8 inhibition alter ECM-related gene or protein expression?

Author Response: We appreciate the reviewer’s insight regarding the role of ECM in leiomyoma growth. While ECM remodeling is indeed a key feature of leiomyoma biology, this study focused primarily on cell proliferation and apoptosis. Therefore, we did not assess ECM-related gene or protein expression. Future investigations—particularly in vivo—will be essential to clarify how CDK8 inhibition may influence ECM dynamics and tissue architecture (page 21, lines 452–455).

8. Progesterone receptor (PR) plays a key role in leiomyoma cell survival. Was PR expression examined?

Author Response: We appreciate the reviewer’s comment regarding PR signaling in leiomyoma biology. While PR is known to contribute to leiomyoma cell survival, our study focused on CDK8 inhibition and its downstream effects, particularly in relation to estrogen receptor (ER) signaling, which has been previously linked to CDK8 activity. In our culture system, both estradiol and progesterone were included to mimic the hormonal environment of the secretory phase. However, we did not assess PR expression or activity in this study. Future investigations may explore whether CDK8 modulation interacts with PR signaling pathways, especially in MED12 WT and MUT contexts. This limitation has been acknowledged in the revised manuscript (page 21, lines 455–459).

9. In Figures 2, 3, and 4, only two-way ANOVA is reported. Post hoc analyses are necessary to clarify which specific group comparisons are statistically significant.

Author Response: We thank the reviewer for highlighting the need for post hoc analyses. In our original submission, we primarily reported the results of two-way ANOVA. As the reviewer suggested, we carefully re-examined our analyses. Most datasets showed significant main effects of Senexin B and EP without significant interaction. Therefore, we considered that reporting the main effects was the most appropriate approach. However, we found one dataset in which a significant interaction between Senexin B and EP had been overlooked. For this dataset, we have now performed Tukey’s post hoc test to clarify the specific group differences, and the corresponding figure has been revised accordingly (Fig. 5B). The corresponding changes have been made to the revised manuscript, and the statistical approach is now described more explicitly in Materials and Methods (page 10, lines 213–217) and figure legends (Figs. 2–6).

Recommendation: Major Revision This is a valuable study with potential clinical implications. However, substantial revisions are required to address experimental limitations and refine the mechanistic interpretation.

Reviewer #2

The authors evaluated the impact of MED12 mutation and CDK8 activity on uterine leiomyoma growth and response to Gn-RH agonist treatment. The results are interesting and informative. The manuscript was well written. There were no points to be revised.

Author Response: We sincerely thank the reviewer for the positive evaluation and supportive comments on our manuscript.

Reviewer #3 This manuscript reports on MED12 mutation and CDK8 activity in relation to uterine leiomyoma growth. Overall, the manuscript is well written and the findings are potentially important. However, several essential details and interpretations are missing. To improve the manuscript for publication, I have the following suggestions:

Author Response: We thank the reviewer for the constructive feedback and recognition of the manuscript’s clarity and significance. In response, we conducted additional experiments, refined mechanistic interpretations, and revised relevant sections accordingly. Specific responses to each comment are provided below.

1. The authors describe the relationship between ERα expression and CDK8 activity in primary cultured cells. However, there is no investigation of steroid receptor expression in surgical specimens. To better clarify the impact of altered CDK8 activity on estrogen responsiveness, I recommend including data, if available, on the expression of ERα or estrogen-responsive genes such as GREB1 in the surgical samples.

Author Response: As per the reviewer’s suggestion, we evaluated ERα expression in surgical specimens using immunohistochemistry (new data is shown in S2 Fig) and found no significant correlation between ERα and CDK8 activity. We have also acknowledged this limitation in the discussion, noting that we did not take the menstrual cycle into account (Materials and Methods, page 8, lines 157–169; Discussion, page 20, lines 421–425).

2. In the cell culture experiments, both estradiol (E2) and progesterone (P4) were used. The rationale for adding P4, in addition to E2, should be clearly explained. Moreover, since the authors discuss the involvement of the estrogen receptor in regulating cell proliferation, they should clarify how the potential proliferative effects of P4 were excluded in their interpretation.

Author Response: We have clarified that we established EP (+/–) conditions to reproduce the low EP environment induced by GnRH agonist therapy in the manuscript (page 19, lines 405–407). Since recent studies have highlighted the effects of P4 on leiomyomas, we also modeled the hormonal environment of the secretory phase, in which both E2 and P4 are present, to better reflect in vivo conditions.

3. P14, L.304–306: “CDK8 inhibition in primary cultured MED12 WT cells increased the amount of CDK8 immunoprecipitated with anti-MED12 antibody and decreased CDK8 kinase activity.” However, there are no experimental data on CDK8 kinase activity in the primary cultured cells to support this conclusion. The authors should either provide the supporting data or revise the sentence accordingly.

Author Response: The experimental data supporting this conclusion are shown in Fig. 5B. Specifically, CDK8 kinase activity was evaluated as CTD-pS5 phosphorylation by immunoprecipitated CDK8 (IP-CDK8), and the amount of IP-CDK8 reflected the CDK8 immunoprecipitated with anti-MED12 antibody. As described in the Results, CTD-pS5 levels were significantly reduced 4 days after Senexin B treatment (LM1: p = 0.01, LM2: p < 0.001), whereas IP-CDK8 levels were significantly increased in LM2 and tended to increase in LM1 (LM1: p = 0.24, LM2: p < 0.001) (page 16, lines 330–333). To clarify this point, we have revised the manuscript to explicitly state that CDK8 activity was assessed as CTD-pS5 phosphorylation, and that IP-CDK8 reflects the amount of CDK8 immunoprecipitated with anti-MED12 antibody (page 18, lines 380–382).

We believe that the manuscript has been significantly improved based on these revisions and thank you again for your constructive criticism. We look forward to hearing from you soon.

---

## [Decision Letter · Decision Letter 1]

24 Nov 2025

Impact of *MED12* mutation and CDK8 activity on uterine leiomyoma growth and response to gonadotropin-releasing hormone agonist treatment

PONE-D-25-24477R1

Dear Dr. Asano,

We’re pleased to inform you that your manuscript has been judged scientifically suitable for publication and will be formally accepted for publication once it meets all outstanding technical requirements.

Kind regards,

Kazunori Nagasaka

Academic Editor

PLOS ONE

Additional Editor Comments (optional):

Dear Authors,

Thank you very much for submitting the revised version of your manuscript.

I have carefully reviewed your detailed responses as well as the revised text, figures, and supplementary materials.

I am pleased to confirm that all reviewer comments have been thoroughly and satisfactorily addressed.

The authors have provided additional experiments, clarified methodological points, improved figure quality, and substantially strengthened the mechanistic interpretation.

These revisions have markedly improved the clarity, rigor, and overall impact of the work.

Both Reviewer #1 and Reviewer #3 indicated that their concerns have now been fully resolved.

Reviewer #2 had no concerns in the first round and remains supportive of publication.

The new data and explanations convincingly address all previously raised issues.

Based on the comprehensive revisions and the reviewers’ positive evaluations, I am pleased to recommend the manuscript for acceptance in PLOS ONE.

I would like to express my great appreciation to the authors for their thoughtful and diligent revisions, and for the care taken in addressing each comment in depth.

Congratulations, and thank you for choosing PLOS ONE for the publication of your work!

Sincerely,

Kazunori Nagasaka

Plos One Editorial Office

Reviewers' comments:

Reviewer's Responses to Questions

**Comments to the Author**

Reviewer #1: All comments have been addressed

Reviewer #3: (No Response)

2. Is the manuscript technically sound, and do the data support the conclusions?

Reviewer #1: Yes

Reviewer #3: Yes

3. Has the statistical analysis been performed appropriately and rigorously?

Reviewer #1: Yes

Reviewer #3: Yes

4. Have the authors made all data underlying the findings in their manuscript fully available?

Reviewer #1: Yes

Reviewer #3: Yes

5. Is the manuscript presented in an intelligible fashion and written in standard English?

Reviewer #1: Yes

Reviewer #3: Yes

Reviewer #1: Thank you for submitting the revised version of your manuscript.I have carefully reviewed your responses and the revised text. All of the reviewers’ comments have been thoroughly addressed, and the manuscript has undergone significant improvement.I am pleased to recommend acceptance of the paper for publication.

I would like to express my appreciation to the authors for their thoughtful revisions and efforts.

Reviewer #3: The authors have adequately revised the manuscript in response to my comments.

The new data and clarifications sufficiently resolve all concerns.

I recommend acceptance of the revised version.

**Do you want your identity to be public for this peer review?** For information about this choice, including consent withdrawal, please see our Privacy Policy

Reviewer #1: No

Reviewer #3: No

---

## [Editor Report · Acceptance letter]

PONE-D-25-24477R1

PLOS One

Dear Dr. Asano,

I'm pleased to inform you that your manuscript has been deemed suitable for publication in PLOS One. Congratulations! Your manuscript is now being handed over to our production team.

Kind regards,

on behalf of

Professor Kazunori Nagasaka

Academic Editor

PLOS One